

# Impact of emotional intelligence on adherence to the Mediterranean diet in elementary education school students. A structural equation model

Eduardo Melguizo-Ibáñez[1], Georgian Badicu[2], Filipe Manuel Clemente[3,4,5], Ana Filipa Silva[3,4], Jose Luis Ubago-Jiménez[1] and Gabriel González-Valero[6]

[1] Department of Didactics of Musical, Plastic and Corporal Expression, Faculty of Education Sciences, University of Granada, Granada, Spain
[2] Department of Physical Education and Special Motricity, Transilvania University of Brasov, Brasov, Romania
[3] Escola Superior Desporto e Lazer, Instituto Politécnico de Viana do Castelo, Rua Escola Industrial e Comercial de Nun'Álvares, Viana do Cas, Viana do Castelo, Portugal
[4] Research Center in Sports Performance, Recreation, Innovation and Technology (SPRINT), Melgaço, Portugal
[5] Instituto de Telecomunicações, Delegação da Covilhã, Lisboa, Portugal
[6] Department of Didactics of Musical, Plastic and Corporal Expression, Melilla Campus, University of Granada, Granada, Spain

Corresponding author
Georgian Badicu,
georgian.badicu@unitbv.ro

## ABSTRACT

**Background**. Adolescence is one of the stages where a large number of physical, psychological and emotional changes occur, the latter playing a key role in adherence to a healthy dietary pattern. Taking into account the above, this research reflects the objectives of developing an explanatory model of the incidence of attention, clarity and emotional repair on adherence to the Mediterranean diet and to contrast the structural model through a multigroup analysis based on Mediterranean adherence to this dietary pattern.
**Methods**. To this end, a comparative descriptive cross-sectional study has been carried out with 293 students from the third cycle of primary education. The instruments used were a sociodemographic questionnaire, the KIDMED test and the Trait Meta-Mood Scale (TMMS-24).
**Results**. Students who show low adherence to the Mediterranean diet have a negative association with the repair of negative emotions. Participants who show a medium adherence to the Mediterranean diet, it is observed that they have a negative relationship with emotional clarity, while students who claim to have a high adherence to the Mediterranean diet show positive relationships with each of the constructs that make up emotional intelligence.

## INTRODUCTION

Currently, modernisation and cultural changes in Western societies are bringing sociological changes that have a negative impact on physical and psychological development in adolescence (*Joensuu et al., 2021*; *Beghin et al., 2021*). This stage of human development
occurs between childhood and adulthood, playing a key role not only because of the physical changes that occur, but also because of the psychological and social changes that it entails (*Corder et al., 2019*). This phase, being a period of so many changes, increases the vulnerability of subjects to the acquisition of harmful lifestyle habits, making them a risk group (*Bornhost et al., 2020*).

Studies point to the progressive abandonment of a dietary pattern in adolescence, as young people begin to have greater control over their dietary pattern and a greater shift towards Western diets based on calorie imbalance (*Appanah et al., 2021*; *Appanah et al., 2020*) leading to a progressive deterioration of optimal adherence to the Mediterranean diet (*Wilson et al., 2020*). This dietary pattern is characterised by the intake of a wide variety of foods including whole grains, olive oil, bread, dairy products, fruits, vegetables and nuts (*Martínez-González, Gea & Canela, 2019*). At the same time, *Muros et al. (2017)* point out that positive adherence to the Mediterranean diet brings numerous health benefits such as reduced waist circumference, increased life expectancy and adequate body fat percentage, reduced likelihood of cardiovascular and neurodegenerative diseases (*Ubago-Jiménez et al., 2020*) and benefits in emotional control (*Ferrer-Cascales et al., 2019*).

Emotional competence is a fundamental element in the day-to-day development of students, as it equips them with the necessary skills to cope with various negative situations emotionally, as well as to promote psychological well-being (*Puertas-Molero et al., 2020*). Being emotionally competent brings benefits such as improved concentration, control over stressful situations and improved personal self-motivation (*Castro-Sánchez, Zurita-Ortega & Chacón-Cuberos, 2018*). This is why emotional intelligence is conceived as a construct that constitutes the psychological development of emotions (*Puertas-Molero et al., 2020*), being composed of attention to these states, clarity of understanding and repair in the face of negative emotions (*Mayer, Caruso & Salovey, 1999*), these constructs playing a key role in academic success and mental, social and educational well-being (*Puertas-Molero et al., 2020*).

Numerous studies have attempted to explain the incidence of the emotional environment on adherence to an active and healthy lifestyle. In this case, the research conducted by *Melguizo-Ibáñez et al. (2020)* found that adolescents who show better emotional control show better adherence to the Mediterranean diet, affirming (*Toet et al., 2022*) disruptive states, tend to carry out a process of overeating, characterized by the the presence of emotional eaters, who, when faced with negative emotions or negative intake of unhealthy foods such as pastries and an increase in alcoholic beverages (*Chang et al., 2021*), with adolescents being a population at risk due to the numerous emotional changes that occur during this stage (*Melguizo-Ibáñez et al., 2021*).

Therefore, focusing attention on the above, the present study reflects the objectives of (a) developing an explanatory model of the impact of attention, clarity and emotional repair on adherence to the Mediterranean diet and (b) contrasting the structural model by means of a multi-group analysis in terms of Mediterranean adherence to this dietary pattern.

Finally, the following research hypothesis is proposed:

Participants who show high adherence to the Mediterranean diet will obtain better associations between said dietary pattern and the tridimensional construct that composes emotional intelligence than those who show low or medium adherence.

Then, in light of the literature reviewed and the different relationships expected between variables, an exploratory model of adherence to Mediterranean diet in children will be explored (Fig. 1).

## MATERIALS & METHODS

### Design and participants

The present study reflects a non-experimental design (ex post facto), descriptive and cross-sectional. The sample consisted of schoolchildren in the third cycle of primary education in the province of Granada, whose ages ranged between 11 and 12 years (11.47 ± 0.32). Likewise, convenience sampling was used, inviting those young people who met the requirements of the present study to collaborate. In this case, the inclusion criteria were the following: being in the third cycle of primary education (fifth and sixth grade), being an exclusion criterion not to be in this educational stage. The sample consisted of a total of 293 students, of whom 147 were male (50.2%) and 146 female (49.8%).

### Variables and instruments

Sociodemographic questionnaire which was designed to collect variables such as sex (male and female) and age.

*Kidmed questionnaire* developed by *Serrá-Majem et al. (2004)*. This instrument is made up of a total of 16 items, which are answered positively or negatively. Items 5, 11, 13 and 15 have a negative character and if answered positively they are valued with −1 point. The values of adherence to the Mediterranean diet are as follows: optimal diet (≥8 points), needs improvement (2–7 points) and low diet quality (≤ 1 point). For the present study, Cronbach's alpha obtained a score of $\alpha = 0.889$.

*Trait Meta-Mood Scale (TMMS-24)* developed by *Salovey et al. (1995)*, but for the present research, the version adapted to Spanish by *Fernández-Berrocal, Extremera & Ramos (2004)* has been used. This questionnaire assesses emotional intelligence as a trifactorial construct, where emotional attention, emotional clarity and emotional repair are assessed through a summative measure. In the present study, emotional attention obtained an $\alpha = 0.843$, emotional clarity an $\alpha = 0.842$ and finally, emotional repair an $\alpha = 0.814$.

### Procedure

The procedure has been similar to that developed in other studies (*Melguizo-Ibáñez et al., 2022a*; *Melguizo-Ibáñez et al., 2022b*). The first step was to carry out a bibliographical search of the current problem in order to find out the state of the art. Subsequently, the Department of Didactics of Musical, Plastic and Corporal Expression of the University of Granada contacted various educational centres by telematic means, informing them of the purpose of the study, and once a favourable response was obtained, an email was sent to the students' legal guardians, informing them that the data would be treated anonymously
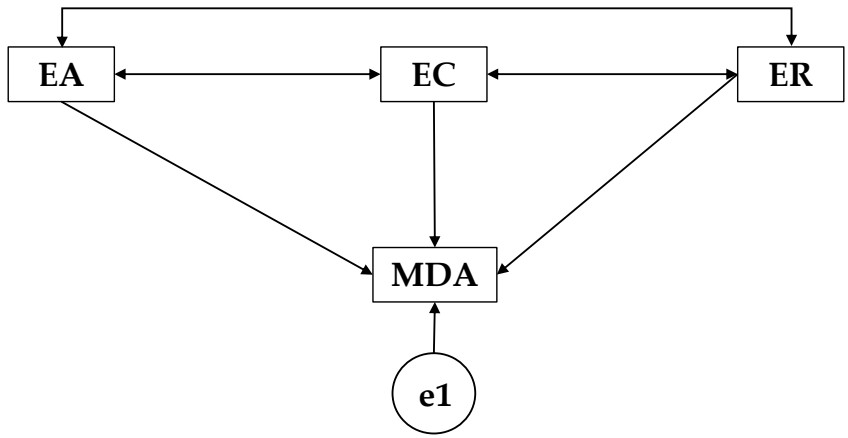

**Figure 1  Proposed theoretical model.** Emotional Attention (EA); emotional Clarity (EC); emotional Repair (ER); Mediterranean Diet Adherence (MDA).

and exclusively for scientific purposes. Once written consent had been obtained from the legal guardians for their children to participate in the research, a Google Form was sent to the different schools, where the participants responded to the instruments described above. Because the investigators were unable to access the various schools due to the COVID-19 pandemic, teachers at the various schools were instructed to resolve any questions that arose. This research followed the ethical principles established in the Declaration of Helsinki and was approved and supervised by an ethics committee of the University of Granada (1230/CEIH/2020). To ensure that participants did not respond randomly to the questions, two questionnaires were duplicated in order to ensure that they were not randomly answered, however, a total of 25 questionnaires had to be eliminated due to inadequate completion.

## Data analysis

The IBM SPSS Statics 25.0 statistical programme (IBM Corp, Armonk, NY, USA) was used to analyse the data, where an analysis of frequencies and means was carried out, using Cronbach's Alpha to determine the internal consistency of the instruments, establishing the reliability index at 95%. To study the normality of the sample, the Kolmogorov–Smirnov purity test was used, obtaining normal normality.

For the elaboration of the structural equation models, the IBM SPSS Amos 26.0 program (IBM Corp., Amonk, NY, USA) was used to establish the relationships between the variables that make up the theoretical model (Fig. 1). A general model was developed for the total sample and subsequently a model was presented according to the degree of adherence to the Mediterranean diet (low, medium and high). Likewise, each model is made up of three exogenous variables (EA, EC, ER) and one endogenous variable (MDA). In the case of the latter variable, a causal explanation has been made taking into account the observed associations between the indicators and the reliability of measurement, so that the measurement error of the observed variables has been included in the model. Likewise,

the unidirectional arrows represent lines of influence between the latent variables and are interpreted from regression weights. A significance level of 0.05 was established using Pearson's Chi-Square test.

In this case, adherence to the Mediterranean diet acts as an endogenous variable, which is affected by the three-factor construct of emotional intelligence (attention, clarity and repair).

Finally, the fit of the model was evaluated after estimating the different parameters of the model. According to the criteria established by *McDonald & Marsh (1990)* and *Bentler (1990)*, the goodness of fit must be assessed on the Chi-Square fit, whose values associated with p and not significant indicate a good fit of the model. In this case, the comparative fit index (CFI; values above 0.95 indicate a good model fit), the goodness-of-fit index (GFI; values above 0.90 indicate an acceptable fit), the incremental reliability index (IFI; values above 0.90 indicate an acceptable fit) and the root mean square approximation (RMSEA; values below 0.1 indicate an acceptable model fit).

## RESULTS

Table 1 shows the sociodemographic characteristics of each model developed. In this case it is observed that the total sample is made up of 147 boys and 146 girls. It should also be noted that 93 participants showed an optimal adherence to the Mediterranean diet, 166 showed an average adherence and 34 showed a low adherence to this dietary pattern.

The model develop for low adherence showed a good fit for each of the indices. The Chi-square showed a significant $p$-value ($X^2 = 77.699$; $df = 16$; pl = 0.000). In this case, such data cannot be interpreted in an independent way due to the sample size and susceptibility of the sample, therefore other standardized fit indices have been used (*Tenenbaum & Eklund, 2007*). The comparative fit index (CFI) obtained a value of 0.949, the normalized fit index (NFI) showed a value of 0.925, the incremental fit index (IFI) obtained a score of 0.958 and the Tucker Lewis index (TLI) evidenced a value of 0.953. Finally RMSEA obtained a value below 0.1, namely 0.052.

Figure 2 and Table 2 show a positive relationship between low adherence to the Mediterranean diet (MDA) and emotional attention (EA) ($r = 0.058$), occurring exactly the same with emotional clarity (EC) ($r = 0.074$). On the contrary, a negative relationship was observed with emotional repair (ER) (r = −0.144). Continuing with the variables that make up emotional intelligence, for emotional attention (EA), positive relationships are observed with emotional clarity (EC) ($r = 0.215$) and emotional repair (ER) ($r = 0.377$). Positive relationships were also observed between emotional clarity (EC) and emotional repair (ER) ($r = 0.432$).

The proposed model for average adherence to a healthy dietary pattern obtained good scores for the different indices that evaluate the fit of the model. The Chi-square showed a significant $p$-value ($X^2 = 75.653$; $df = 16$; pl = 0.000). Likewise, the comparative fit index obtained a score of 0.959, the normalized fit analysis (NFI) showed a score of 0.952, these being excellent values. In addition, the incremental fit index (IFI) was 0.945 and the Tucker-Lewis index (TLI) obtained a value of 0.944. Finally, the root mean square error of approximation analysis (RMSEA) obtained a score of 0.050.

**Table 1  Sociodemographic characteristics of the different proposed models.**

| | Mediterranean Diet Adherence | | | Total |
|---|---|---|---|---|
| | High Adherence | Medium Adherence | Low Adherence | |
| Male | 51 (54.8%) | 76 (45.8%) | 20 (58.8%) | 147 (50.2%) |
| Female | 42 (45.2%) | 90 (54.2%) | 14 (41.2%) | 146 (49.8%) |
| Total | 93 (100.0%) | 166 (100.0%) | 34 (100.0%) | 293 (100.0%) |

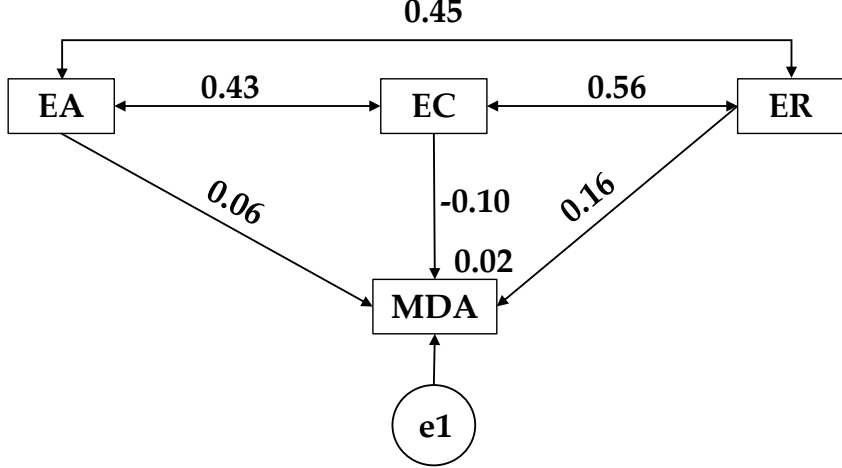

**Figure 2  Theoretical model proposed for low adherence to the Mediterranean diet.** Emotional Attention (EA); Emotional Clarity (EC); Emotional Repair (ER); Mediterranean Diet Adherence (MDA).

**Table 2  Structural model for low adherence to Mediterranean diet.**

| Associations between variables | R.W. | | | | S.R.W. |
|---|---|---|---|---|---|
| | Estimates | S.E. | C.R. | p | Estimates |
| MDA←EA | 0.008 | 0.025 | 0.313 | 0.754 | 0.058 |
| MDA←EC | 0.011 | 0.029 | 0.386 | 0.700 | 0.074 |
| MDA←ER | −0.019 | 0.027 | −0.711 | *** | −0.144 |
| EC←→EA | 0.105 | 0.087 | 1.205 | 0.228 | 0.215 |
| EC←→ER | 0.210 | 0.092 | 2.278 | 0.023 | 0.432 |
| EA←→ER | 0.205 | 0.101 | 2.024 | 0.043 | 0.377 |

**Notes.**

Regression weights (R.W); Standardized regression weights (S.R.W); Estimation error (S.E); Critical ratio (C.R).

Emotional Attention (EA); Emotional Clarity (EC); Emotional Repair (ER); Mediterranean Diet Adherence (MDA).

*** $p < 0.001$.

**Table 3  Structural model for medium adherence to the Mediterranean diet.**

| Associations between variables | R.W. | | | | S.R.W. |
|---|---|---|---|---|---|
| | Estimates | S.E. | C.R. | $p$ | Estimates |
| MDA←EA | 0.005 | 0.007 | 0.667 | 0.505 | 0.059 |
| MDA←EC | −0.010 | 0.010 | −1.051 | 0.293 | −0.101 |
| MDA←ER | 0.015 | 0.009 | 1.638 | 0.101 | 0.158 |
| EC←→EA | 0.285 | 0.056 | 5.071 | *** | 0.431 |
| EC←→ER | 0.299 | 0.048 | 5.071 | *** | 0.556 |
| EA←→ER | 0.302 | 0.058 | 5.240 | *** | 0.448 |

**Notes.**

Regression weights (R.W); Standardized regression weights (S.R.W); Estimation error (S.E); Critical ratio (C.R).

Note 2: Emotional Attention (EA); Emotional Clarity (EC); Emotional Repair (ER); Mediterranean Diet Adherence (MDA).

*** $p < 0.001$.

Table 3 and Fig. 3, evidence a positive relationship between adherence to the Mediterranean diet (MDA) and emotional attention (EA) ($r = 0.059$), the same pattern being observed with emotional repair (ER) ($r = 0.158$). In contrast, a negative relationship was shown between emotional clarity (EC) and adherence to a healthy dietary pattern (MDA) ($r = −0.101$). Continuing with the relationships found between the different emotional constructs, positive relationships are shown between emotional clarity (EC) and emotional attention (EA) ($p < 0.001; r = 0.431$). A positive relationship was also observed between emotional clarity (EC) and emotional repair (ER) ($p < 0.001; r = 0.556$). Finally, a positive relationship was obtained between attention to feelings (EA) and emotional repair (ER) ($p < 0.001; r = 0.448$).

Finally, focusing on the model developed for students showing high adherence to the Mediterranean diet, it showed good scores for each of the indices. The Chi-square showed a significant $p$-value ($X^2 = 74.345; df = 16; \text{pl} = 0.000$). The comparative fit index (CFI) analysis obtained a value of 0.959, which represents an excellent score. The normalised fit index (NFI) analysis obtained a value of 0.952, the incremental fit index (IFI) was 0.9945 and the Tucker-Lewis index (TLI) obtained a value of 0.944, all of which were excellent. In addition, the root mean square error of approximation analysis (RMSEA) also obtained a value of 0.051.

Looking at the results shown in Fig. 4 and Table 4, there is a positive relationship between high adherence to the Mediterranean diet (MDA) and emotional attention (EA) ($r = 0.049$), emotional clarity (EC) ($r = 0.186$) and emotional repair ($r = 0.015$). Likewise, among the emotional constructs, positive relationships were obtained between 'emotional clarity (EC) and emotional attention (EC) ($p < 0.001; r = 0.391$) and emotional repair (ER) ($p < 0.001; r = 0.607$), with another positive relationship between emotional attention (EC) and emotional repair (ER) ($r = 0.340$).

## DISCUSSIONS

The present study reflects the influence of emotions on adherence to a healthy dietary pattern, in this case to the Mediterranean diet in adolescent students in the third cycle of primary education. In this case the platen research hypothesis is fulfilled, as the model
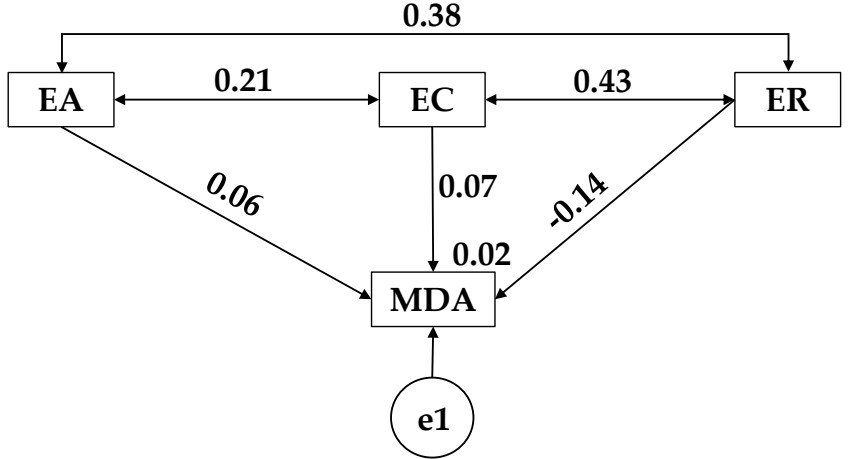

**Figure 3** **Theoretical model proposed for medium adherence to the Mediterranean diet.** Emotional Attention (EA); Emotional Clarity (EC); Emotional Repair (ER); Mediterranean Diet Adherence (MDA).

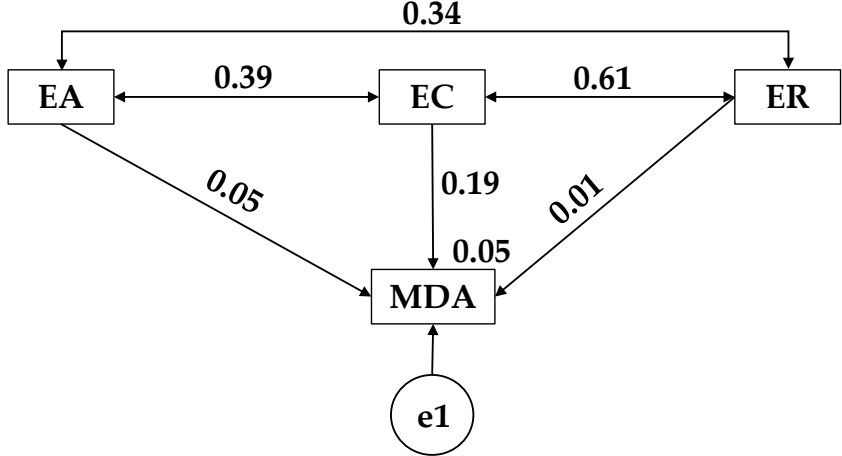

**Figure 4** **Theoretical model proposed for high adherence to the Mediterranean diet.** Emotional Attention (EA); Emotional Clarity (EC); Emotional Repair (ER); Mediterranean Diet Adherence (MDA).

equations reveal that having a low adherence to the Mediterranean diet reports worse associations between the emotional and dietary domains. The results obtained respond to the proposed objectives, which is why this discussion aims to compare the results obtained in this study with those obtained in other similar studies.

Students who show low adherence to the Mediterranean diet, a negative relationship is observed between emotional repair and positive adherence to a healthy dietary pattern. Results very similar to those obtained in the present research were concluded by *Herren et al. (2021)* and *Melguizo-Ibáñez et al. (2022a)*; *Melguizo-Ibáñez et al. (2022b)*, where *Trigueros et al. (2020)* affirm the existence of a process of emotional eating. In this case, this process consists of overeating unhealthy foods, such as pastries and sweets in times of stress or

**Table 4  Structural model for high adherence to the Mediterranean diet.**

| Associations between variables | R.W. | | | | S.R.W. |
|---|---|---|---|---|---|
| | Estimates | S.E. | C.R. | *p* | Estimates |
| MDA←EA | 0.005 | 0.011 | 0.436 | 0.663 | 0.049 |
| MDA←EC | 0.020 | 0.014 | 1.405 | 0.160 | 0.186 |
| MDA←ER | 0.002 | 0.016 | 0.113 | 0.910 | 0.015 |
| EC←→EA | 0.268 | 0.077 | 3.491 | *** | 0.391 |
| EC←→ER | 0.323 | 0.065 | 4.980 | *** | 0.607 |
| EA←→ER | 0.198 | 0.064 | 3.084 | 0.002 | 0.340 |

**Notes.**

Regression weights (R.W); Standardized regression weights (S.R.W); Estimation error (S.E); Critical ratio (C.R).

Emotional Attention (EA); Emotional Clarity (EC); Emotional Repair (ER); Mediterranean Diet Adherence (MDA).

[***]$p < 0.001$.

anxiety or negative emotions. In this case, research by Linswiler et al. (1989) states that people who mismanage their emotions tend to experience states such as anxiety and stress, leading to a process of overeating. This may also be due to the fact that students may have received little emotional training, and are able to overcome such states through the process described above (*Sabingoz & Dogan, 2019*).

Continuing with the model developed for participants showing a medium adherence to the Mediterranean diet, a negative relationship with emotional clarity is denoted. Very distant results were obtained by *Marchena, Bernabéu & Iglesias (2020)*, where *Melguizo-Ibáñez et al. (2021)* affirming that emotional clarity plays a key role in knowing the different emotional states that are experienced and whether they are beneficial or detrimental. Likewise, emotional clarity plays a key role with emotion repair as students need to understand whether a certain state makes them feel good or bad and thus recover from it if it harms them without engaging in excessive intake of unhealthy foods (*Trigueros et al., 2020*).

For the model developed for students who show a high adherence to the Mediterranean diet, it is observed that there is a positive relationship with the three constructs that comprise emotional intelligence. Very similar results were obtained by *Marchena, Bernabéu & Iglesias (2020)* where they state that a healthy diet brings benefits on different personal dimensions, among which emotional intelligence stands out (*Ubago-Jiménez et al., 2020*).

Such behavior can be explained by the fact that an optimal adherence to the Mediterranean diet brings benefits on the physical and emotional self-concept, generating positive and satisfactory emotions in the physical and mental image of individuals (*Zurita-Ortega et al., 2018*; *Melguizo-Ibáñez et al., 2022a*; *Melguizo-Ibáñez et al., 2022b*).

Regarding the implications that this research offers in the field of health, it is the presence and importance of emotions when it comes to adherence to a healthy dietary pattern. Likewise, the presence of negative emotions together with disruptive process that is detrimental to the physical and mental health of individuals. Several emotional states can originate a process of emotional overeating, resulting in a practical applications of this research can be extrapolated to the educational field, through an intervention

program focused on emotional control and its impact on an active and healthy lifestyle (*Ubago-Jiménez et al., 2020*; *Melguizo-Ibáñez et al., 2022a*; *Melguizo-Ibáñez et al., 2022b*).

## Limitations and future prospects

The present study reflects a series of limitations, which are described below. The first one is that since this is a cross-sectional study, it is only possible to establish the causal relationships of the variables at that time. In addition, the sample is made up of students from a very specific geographic area, so that it is not possible to establish generalizations in a wider area of the national geography. Likewise, the study is exploratory in nature, requiring a future intervention to better study the relationships.

Focusing attention on future perspectives, it has been proposed to develop a longitudinal intervention program, where, through emotional education, the effects of these states on food intake can be studied.

## CONCLUSIONS

The present study shows that attention, clarity and emotional repair influence positive adherence to a healthy pattern. Based on the different models of equations developed, it is observed that students who show low adherence to the Mediterranean diet have a negative association with the repair of negative emotions. Likewise, continuing with those participants who show a medium adherence to the Mediterranean diet, it is observed that they have a negative relationship with emotional clarity, while students who claim to have a high adherence to the Mediterranean diet show positive relationships with each of the constructs that make up emotional intelligence. Finally, the data from the present research affirm that positive adherence to a healthy dietary pattern results in better understanding, clarity and repair of emotions.

### Funding
The authors received no funding for this work.

### Competing Interests
Georgian Badicu and Filipe Manuel Clemente are Academic Editors for PeerJ.

### Author Contributions
- Eduardo Melguizo-Ibáñez conceived and designed the experiments, performed the experiments, prepared figures and/or tables, and approved the final draft.
- Georgian Badicu analyzed the data, authored or reviewed drafts of the article, and approved the final draft.
- Filipe Manuel Clemente analyzed the data, authored or reviewed drafts of the article, and approved the final draft.
- Ana Filipa Silva analyzed the data, authored or reviewed drafts of the article, and approved the final draft.

- Jose Luis Ubago-Jiménez conceived and designed the experiments, analyzed the data, prepared figures and/or tables, and approved the final draft.
- Gabriel Gonzalez Valero conceived and designed the experiments, performed the experiments, prepared figures and/or tables, and approved the final draft.

## Human Ethics

The following information was supplied relating to ethical approvals (i.e., approving body and any reference numbers):

This research followed the ethical principles established in the Declaration of Helsinki and was approved and supervised by an ethics committee of the University of Granada (1230/CEIH/2020).

## Data Availability

Raw data is available as a Supplementary File.

## Supplemental Information

Supplemental information for this article can be found online at http://dx.doi.org/10.7717/peerj.13839#supplemental-information.

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
