# Peer review of "Impact of emotional intelligence on adherence to the Mediterranean diet in elementary education school students. A structural equation model"

_PeerJ, doi:10.7717/peerj.13839_

## Round 0.1 · original submission · Major Revisions

Dear authors,

Please respond to reviewers 'comments point by point, follow reviewers' suggestions carefully, and make appropriate revisions.

·

Basic reporting

Thank you for the opportunity to review this paper. While the topic is of interest in its current form it will require more work before publication. There are a number of areas that require rewriting or clarification. I will comment on these areas section by section.
ABSTRACT:
Abstract doesn't summarize the aim, methodology and conclusion clearly. Suggest it is re-written.

Introduction.
The introduction is easy to read, however did not extend existing knowledge on this topic. It should include more update references regarding the association among emotional intelligence, diet, and health.
After the purpose statement, please provide a hypothesis for what the authors think the results will yield.

Experimental design

Methods
Some important information appears to be presently omitted from the methods section. Further description of the sampling procedure would be helpful for the reader. The analysis process is a bit unclear.

Validity of the findings

Some important information also appears to be presently omitted from the methods and results section. Have you tested the reliability of your data? If yes, please include the results.

Additional comments

Discussion
In general, the first paragraph of the discussion should at least state which hypotheses were supported. Then the authors should follow with how their results compare with similar data, and what the authors results adds to the literature (different / unique aspects of the data). Several points are made in the discussion, but it is not clear to this reviewer how results from the current study are novel or add to the literature.
The authors shortly discuss several possible explanations for the findings. The authors did not discuss what is novel about this research or what it offers in terms of health implications. The authors did not discuss how this research may be disseminated into greater practice. Moreover, the limitations and the strengths of this research were not discussed at all.

Reviewer 2 ·

Basic reporting

No comment

Experimental design

No comment

Validity of the findings

No comment

Additional comments

First of all, I would like to thank the opportunity to review this exciting and well-written article. It is a novel work providing evidence on emotional intelligence's impact on adherence to the Mediterranean diet. It has been demonstrated that the Mediterranean Diet profoundly affects public health, especially in earlier stages, and has gradually disappeared in recent years. However, to be published, this article should review several issues:

1. Regarding the TMMS-24 scale, the authors say that they used a translated questionnaire. Is this adapted questionnaire validated in Spanish?

2. What were the inclusion and exclusion criteria? In lines 74-76, the authors say, "those people who met requirement of study ....." but these criteria are not defined.

3. In lines 211-214, the authors write: "Results very similar to those obtained in the present research were concluded by Herren et al. (2021), where Trigueros et al. (2020) state that at times when negative emotional states are experienced, many adolescents overcome these by overeating an oversaturated intake of unhealthy foods. However, the idea they want to convey is not clear enough. I would ask them to rewrite it, highlighting what they want to say.

4. As in lines 211-214, lines 218-221 are not clear enough. I would also ask the authors to rewrite it, highlighting what they want to say.

5. In a general way, the discussion is a bit weak. The authors should investigate the mechanisms related to how emotional intelligence could influence adherence in more depth.

6. Additionally, two concepts are confused in the text: adherence to the Mediterranean Diet and overfeeding. Please, review the whole text paying attention to this aspect.


I am sure that the publication will improve remarkably once these details are solved.

Reviewer 3 ·

Basic reporting

As the authors are proposing a relationship between emotional attention, emotional clarity, emotional repair, and adherence to the mediterranean diet, I expected to find background about those relationships in the introduction, as well as the different study hypotheses regarding those relationships. Only once all the relationships are exposed, I expected to see something like a sentence that say "Then, in light of the litterature reveiwed and the different relationships expected between variables, an exploratory model of adherence to mediterranean diet in children will be explored (See Figure 1).
Moreover, as ou expected to find different relatinships between variables depending on the degree of adherence to the mediterranean diet of children, the authors should hypothesized each of the expected differences in the introduction.

Experimental design

The decision to perform three different models according to the degree of adherence to the Mediterranean diet in unclear for me. If your interest is to see if the independent variables interact differently with the mediteranean diet adherence depending on the degree of adherence to the mediterranean diet, I think that you must perform a multigroup SEM, and to explore the invariance of the model depending of the degree of the adherence to the mediterranean diet, by including the three groups as moderators. This will allow you (1) to present if the model is well fitted for all the three different samples (low, medium and high degree of adherence), (2) to see if the model is or not invariant in term of the degree of adherence, and (3) in which relationships this potential difference exist depending on the degree of adherence to the mediterranean diet. I highly recomend you to perform this multigroup structural analisis model. This can be done with AMOS, by keeping the model you have used and including the three subsamples as groups in this model and to compute a multigroup analysis.

Please, provide the number of children in each "subsample": how many children show low, medium and high levels of adherence to the mediterranean diet, and then will be included in the different samples for the multigroup analysis? Provide also the sociodemographic data (sex percentage, age mean, etc.) for each one of these subsamples.

In relation to the latter, a problem of the study is the sample size. I really think that doing a SEM analysis with a such small sample and dividing this sample in three groups is a relevant problem, because this suppose very small samples size for each one of your actual three different models, a s well as a very small sample size for the multigroup analisis I suggested to performed with the three different groups. I think that this limitation should be reported, and that the study should be reported as an exploratory report that will need further replication with an higher sample size. Another possibility is not to perform a multigroup analysis, and to perfomr a unique model with all the sample, without comparing the results beween low, medium and high degree of adherence.

Moreover, you are presenting a SEM with cross-sectional data, and then the model have to be presented as an explanatory model and not a predictive model, including in the limitation section the need to replicate the data with a longitudinal design.

I think that Tables 1, 2 and 3 are unnecesary. The standardized weights sould be reported in their respective figure, with the p values in the same figure, as ***, ** or ** depending of the level of the p value, and a note indicating to wich p value correspond each. This will ease the interpretation of the data for readers.

The upper and lower values of the RMSEA should be reported.

More than explaining the relationships for each group separately, I think that the comparation between groups should be more interesting and reported, as well as an attempt to explain why those differences in the relationships are found. Moreover, those differences between the path depending on the degree of adherence to the mediterranean diet should be ensured with the multigroup analysis. The authors should explain why a negative relationship is observed between emotional clarity and adherence in children with a medium adherence degree. Why this could be due. What implication this entails? Can the relationships been really ensured, as maybe the sample size of children with medium degree of adherence is very very low?
Please, try to explain why different relations (some negative and other positive) are found between the same variables depending on the degree of mediterranean diet. Which sense this make? Which implications this entail?

Validity of the findings

In relation to the comments previously done, I think that the conclusion are overstated (no multigroup analysis was performed) and that it will be more convenient to explore and explain the differences between groups (low, medium and high adherence). Again, this need to perform a multigroup analysis.
Moreover, the limitation of the sample size is a very relevant problem, and more if we consider that the sample is divided in three diferent groups. This problem remain if you perform a multigroup analysis, and should be reported. Maybe the more coherent is to perform a unique model (not multigroup) with all the sample.

Additional comments

Accronyms of the variables in Figures and Tables should be change in accordance with their english name.

---

## Round 0.2 · Minor Revisions

Dear Authors,

You have made a major revision which has greatly improved the manuscript.

I would like to ask you further to review only the following points:

1) the authors refer to the fact that the questionnaire they use is validated for Spanish. If this is the case, they should indicate it in the text.

2) In the conclusions, the changes made are not sufficient to explain these mechanisms. The conclusion merely summarizes the results. Please pay attention to this point.
Thank you!

Reviewer 2 ·

Basic reporting

No comment

Experimental design

No comment

Validity of the findings

No comment

Additional comments

I want to thank the authors for their efforts to improve this article. However, I still think some aspects should be enhanced for publication.
First, the authors refer to the fact that the questionnaire they use is validated for Spanish. If this is the case, they should indicate it in the text.
On the other hand, I congratulate the authors for their hard work in modifying the entire results section. However, the lack of clarity in the discussion section remains.
Finally, my most significant concern is in the conclusions. The changes made are not sufficient to explain these mechanisms. The conclusion merely summarizes the results. Please pay attention to this point.

---

## Round 0.3 · accepted · Accept

Dear Authors,

Thank you for following the reviewers' suggestions and for making the appropriate reviews.

The manuscript has improved and is now worthy of being published on PeerJ.